# COUNTERFACTUAL IMAGE NETWORKS

## ABSTRACT

We capitalize on the natural compositional structure of images in order to learn object segmentation with weakly labeled images. The intuition behind our approach is that removing objects from images will yield natural images, however removing random patches will yield unnatural images. We leverage this signal to develop a generative model that decomposes an image into layers, and when all layers are combined, it reconstructs the input image. However, when a layer is removed, the model learns to produce a different image that still looks natural to an adversary, which is possible by removing objects. Experiments and visualizations suggest that this model automatically learns object segmentation on images labeled only by scene better than baselines.

## 1 INTRODUCTION

Visual recognition models demand large amounts of annotated data that is expensive to collect, and this cost is amplified for tasks that require densely labeled data, such as semantic segmentation. In this paper, we develop an approach where object segmentation emerges automatically for images only labeled by scene category.

We capitalize on the natural compositional structure of images to learn object segmentation through counterfactual images. An image is counterfactual if it shows a real scene, except part of it has been removed or changed. To learn to segment, we train a model to generate counterfactual images such that they are perceptually realistic, a task the model can solve by removing objects and filling in the holes. For example, if you fully remove the bed from the scene in Figure 1, the image is still realistic. However, if you only partially remove the bed, the image is not realistic anymore. We use this intuition to automatically learn object segmentation.

We develop a stochastic layered model that decomposes an input image into several layers. We train this model so that when all layers are combined together in some order, it reconstructs the input image. However, we also train the model so that if we randomly permute the layers and remove a layer, the combination still appears perceptually real to an adversary. Consequently, the model learns a layered image decomposition that allows parts of the image to be removed. We show that the model automatically learns to isolate objects in different layers in order to make the output image still appear realistic, a signal we capitalize on for learning to segment.

We present three main experiments to analyze this approach. Firstly, experiments show that our model learns to automatically segment images into objects for some scene categories, with only weakly labeled training data, and our approach outperforms several baselines. Secondly, we show that we use a small amount of densely labeled data with our approach to further improve performance. Finally, visualizations suggest that the model can generate the scene behind objects that it learns to segment, enabling us to remove pictures from a wall or take off the bed sheets.

Our main contribution is to introduce a novel method for object segmentation on data only labeled by scene by capitalizing on natural compositional structures in images. While the focus of this paper is on images, the method is general and could be applied to other signals, such as audio. The remainder of this paper describes this contribution. Section 2 reviews related work. Section 3 present our method to auto-encode images with a layered decomposition, and shows how removing image regions is a useful signal for segmentation. Section 4 shows several experiments for semantic segmentation, and section 5 offers concluding remarks. We plan to release all code, data, and models.

Removing objects yields
natural images

Removing random patches
yields unnatural images

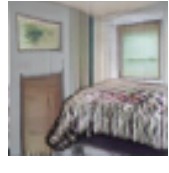 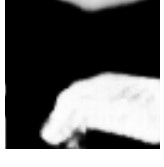 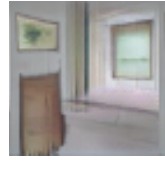   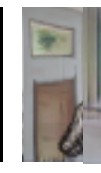 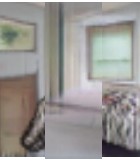

Image                Mask        Removed              Mask        Removed

Figure 1: We make the simple observation that if you remove an object from an image, the image still looks natural (middle). However, if you remove a random patch, the image likely looks unnatural (right). In this paper, we leverage this counterfactual signal for image segmentation.

## 2 RELATED WORK

**Image Segmentation:** Pixel-wise segmentation is widely studied in computer vision. Edge and boundary detection seek to recognize contours between objects (Canny, 1986; Martin et al., 2004; Dollar et al., 2006; Isola et al., 2014), but does not attach category labels to segments. Semantic segmentation instead seeks to both segment objects and assign labels, which is the task that we consider. Tighe & Lazebnik (2011); Yang et al. (2014); Badrinarayanan et al. (2015); Bansal et al. (2016) learn to semantically segment objects in images, however they require large amounts of manual supervision. In this work, we do not require pixel-wise labeled data in order to learn to segment objects; we only require images that are known to be within a certain scene category. In related work, Li & Malik (2016) investigate segmenting objects behind occlusions, but also require supervision. Ehsani et al. (2017) explore how to remove occlusions from images, but require specifying the occlusions a priori. Our work is most related to Sudderth & Jordan (2009), which use layered models for segmentation. However, our work differs because we learn a single model for semantic segmentation that can work across multiple images.

**Layered Visual Models:** Layered image models are widely used in computer vision (Wang & Adelson, 1994; Yang et al., 2012; Sun et al., 2015; Yang et al., 2017; Finn et al., 2016; Vondrick et al., 2016; Huang & Murphy, 2015; Eslami et al., 2016), however here we are leveraging them to segment images without pixel-level human supervision. We develop a model that learns to decompose an image into separate layers, which we use for segmentation. (Yang et al., 2012) is similar to our work in that they generate images by layers, however they do not show that randomly removing layers is a signal for semantic segmentations.

**Noise in Learning:** Dropout (Srivastava et al., 2014) is commonly used in neural networks to regularize training by randomly dropping hidden unit activations. (Huang et al., 2016) also randomly drops neural layers to facilitate training. Our work uses similar mechanism to randomly drop generated layers, but we do it to encourage a semantic decomposition of images into layers of objects. Note that the layers we drop are image layers, not layers of a neural network.

**Emergent Units:** Our work is related to emergent representations for neural networks. For example, recent work shows that hidden units automatically emerge to detect some concepts in visual classification tasks (Zhou et al., 2014) and natural language tasks (Radford et al., 2017). In this work, we also rely on the emergent behavior of deep neural networks, but for object segmentation.

**Unsupervised Representation Learning:** Methods to learn representations from unlabeled data are related but different to this work. For example, spatial context (Doersch et al., 2015) and word context (Mikolov et al., 2013) can be used as supervisory signals for vision and language respectively. While our work is also using unlabeled data within a scene class, we are not learning representations. Rather, we show that object segmentation emerges automatically in our approach.

## 3 COUNTERFACTUAL IMAGE NETWORKS

Our method uses layered models in order to generate counterfactual scenes given an input image. Let $x_i \in \mathbb{R}^{W \times H}$ be an image in our dataset. Note that for simplicity of notation, we assume gray-scale

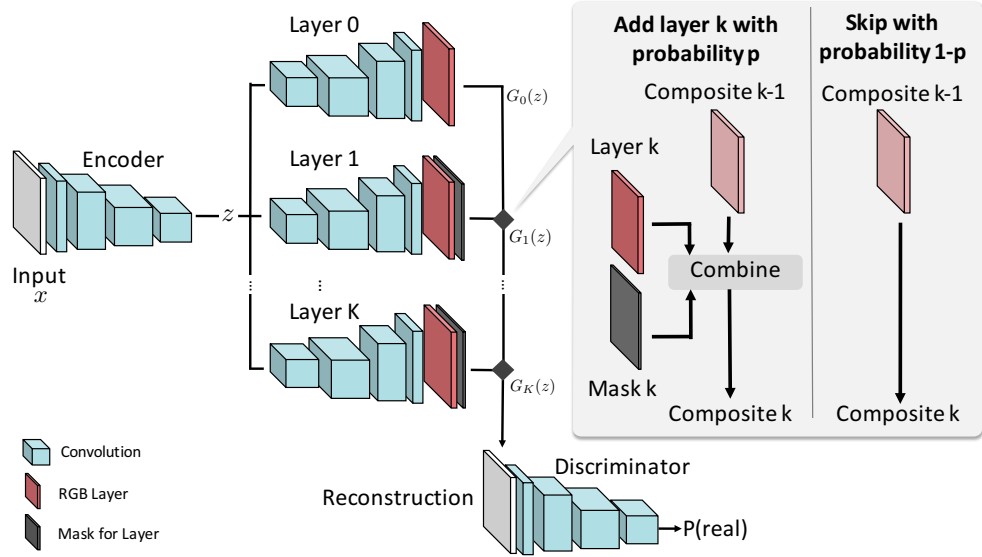

Figure 2: **Network Architecture:** We visualize our neural network architecture. Given an input image, we generate $K$ layers and masks to combine them. However, each layer only has a certain probability of being combined with the current composite.

images, however our method trivially extends to color images. We follow an encoder-decoder setup. We will encode an image into a latent code $z_i \in \mathbb{R}^D$, then decode the code into $K$ image layers.

### 3.1 GENERATION MODEL

We use a simple layered model for image generation. Given a latent code $z \in \mathbb{R}^D$, we stochastically and recursively decode it to produce an image:

$$G_0(z) = f_0(z) \tag{1}$$

$$G_k(z) = \begin{cases} f_{\sigma(k)}(z) \odot m_{\sigma(k)}(z) + G_{k-1}(z) \odot \left(1 - m_{\sigma(k)}(z)\right) & \text{with prob. } p_k \\ G_{k-1}(z) & \text{otherwise} \end{cases} \tag{2}$$

where the $k$th layer is only added with probability $p_k$, and $\sigma$ is a permutation. Our intention is that the neural networks $m_{\sigma(k)}(z) \in \mathbb{R}^{W \times H}$ will generate a mask and $f_{\sigma(k)}(z) \in \mathbb{R}^{W \times H}$ will generate a foreground image to be combined with the previous layer. In our experiments, the the mask $m$ and foreground $f$ networks are shared except for the last layer. To ensure the mask and foreground are in a valid range, we use a sigmoid and tanh activation function respectively. $\odot$ denotes element-wise product. The base case of the recursion, $G_0(z)$, is the background layer. To obtain the final image, we recurse a fixed number of times $K$ to obtain the result $G_K(z)$.

### 3.2 STOCHASTIC COMPOSITION

The generation model $G_K(z)$ is stochastic because each layer is only added with a given probability. We will train $G_K(z)$ to generate images that still look perceptually real to an adversary even when some layers are removed. To be robust to this type of corruption, we hypothesize that the model will learn to place objects in each layer. Removing objects will fool the adversary, however removing an arbitrary patch will not fool the adversary because those images do not occur in nature.

We introduce an additional form of stochasticity by noting that objects in images are not often in the same order with respect to each other. We want the layers to specialize to objects without explicitly enforcing an ordering, so in our model we also randomly permute the foreground layers before disposing and composing to form the counterfactual. Specifically, each foreground layer has an equal probability of being in any position when composing.

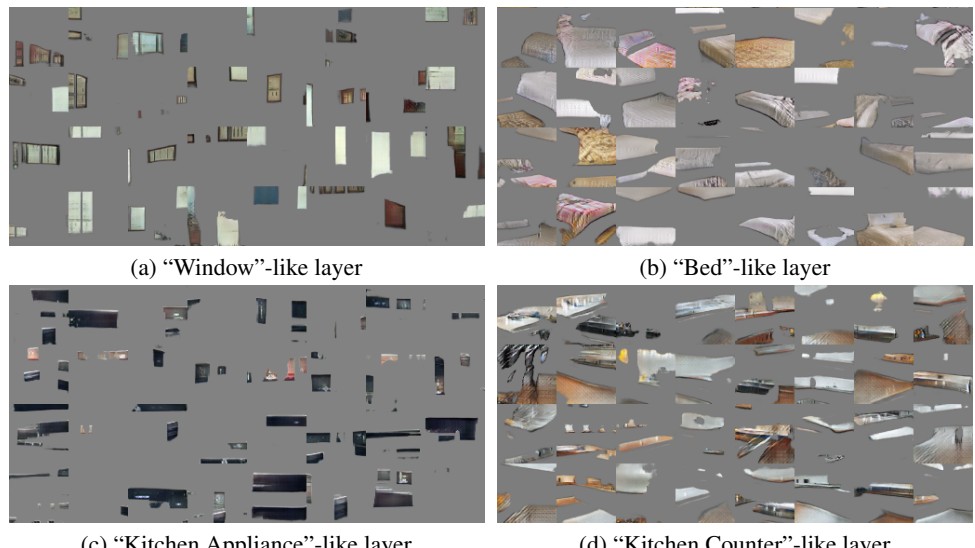

(a) "Window"-like layer

(b) "Bed"-like layer

(c) "Kitchen Appliance"-like layer

(d) "Kitchen Counter"-like layer

Figure 3: **Example Layers:** We visualize some generations from different layers. For example, some layers specialize to segmenting and generating windows, while others specialize to beds. We note that the model does not attach semantics to the layers. Each layer is given a name by hand after training.

We train $G_K(z)$ to generate images that fool an adversary. To do this, we use generative adversarial networks (Goodfellow et al., 2014; Radford et al., 2015). We use a convolutional network $D$ as a discriminator and optimize our generator $G$ to fool $D$ while simultaneously training $D$ to distinguish between generated images and images from the dataset. Figure 3 shows a few qualitative examples of learned layer generations from this model. Notice the network can automatically learn a decomposition of objects and their boundaries.

### 3.3 INFERENCE MODEL

We have so far described the generation process given a latent code $z$. To segment an image $x$, we need to infer this code. We will train an encoder $E(x)$ to predict the code given an image $x$. One possible strategy is to train $E$ to minimize the pixel-wise reconstruction error over our unlabeled dataset, i.e. $\min_E \sum_i \|G_K(E(x_i)) - x_i\|_2^2$. While this will yield better reconstructions, the reconstruction will be low-level and not necessarily semantic.

We therefore use a different strategy. We will train $E$ to reconstruct the latent codes from sampled scenes from the generator, i.e. $\min_E \sum_{z \sim \mathcal{N}(0,I)} \|E(G_K(z)) - z\|_2^2$. While this does not guarantee a strong reconstruction in pixel space, it may enable a more semantic reconstruction, which is our goal. We note this strategy is discussed but not experimented by Dumoulin et al. (2016).

### 3.4 LEARNING

We learn the parameters of the neural networks $D$, $E$, and $G$ jointly. We optimize:

$$\min_{D,E} \sum_{z \sim U} \left[ \log D\left(G_K(z)\right) + \lambda \|E\left(\bar{G}_K(z)\right) - z\|_2^2 \right] + \sum_i \log\left(1 - D\left(x_i\right)\right) \tag{3}$$

$$\max_G \sum_{z \sim U} \log D\left(G_K(z)\right) \tag{4}$$

where $U$ is the uniform distribution on the interval $[-1, 1]$ and $\bar{G}$ indicates that no layers are dropped. To optimize this min-max objective, we alternate between minimizing Equation 3 and maximizing Equation 4 using mini-batch stochastic gradient descent. Note that this objective is similar to a generative adversarial network (Goodfellow et al., 2014), however there is also an encoder $E$. We use $\lambda = 1$. Importantly, to train our model, we only need a collection of unlabeled images within a scene

category. The model will learn to auto-encode images such that layers can be randomly removed and still produce a realistic image.

## 3.5 SEMANTIC SEGMENTATION

We take advantage of the emergent masks of the layers for semantic segmentation. After training, we will have $K$ different masks $m_k(z)$. Since $K$ is typically small (we use $K = 5$), we can manually inspect a few examples on the training set and attach a name to each one. We use these masks as the semantic segmentation prediction. Figure 3 shows a few examples of learned masks from this model.

## 3.6 NETWORK ARCHITECTURE AND IMPLEMENTATION DETAILS

Our network architecture is similar to DCGAN (Radford et al., 2015) when possible. The encoder contains 3 layers of 4x4 convolutions with a stride of 2, followed by a single layer of 3x3 convolutions of stride 1, and then another single layer of 4x4 convolutions of stride 2. Since we use reconstructions for image segmentation, we care about encoding spatial location of the objects, so we use a latent vector of size 64 x 4 x 4. The decoder has identical architecture, but contains up-convolutions instead. Each layer is generated independently from the hidden state vector without tied weights. We add batch normalization (Ioffe & Szegedy, 2015) between layers, leaky ReLU for the encoder and discriminator and ReLU for the generator. We train with Adam (Kingma & Ba, 2014) with learning rate 0.0002 and beta 0.5 for the object discovery experiments and learning rate 0.00002 for finetuning. We train for 2 epochs over the dataset for both scene categories. In all examples we use 5 foreground layers and set the probability that a layer is included to 0.4. We plan on making all code and data publicly available.

## 4 EXPERIMENTS

We present three experiments to evaluate our model. In the quantitative experiments, we evaluate how well layers automatically emerge to classify pixels to belong to a specific object category. In the qualitative experiment, we give examples as to how we can use our layered reconstruction to decompose an image into semantic layers.

## 4.1 EXPERIMENTAL SETUP

We experiment with our approach using images of bedrooms and kitchen scene categories from the LSUN dataset (Yu et al., 2015). For bedrooms, we focus on segmenting bed and window. For kitchens, we focus on segmenting appliances and countertop. The dataset contains a total of $3,033,042$ images of bedrooms and $2,212,277$ images of kitchens which we train separate models on. Note that apart from scene category, these images are unlabeled; they do not have any pixel level annotations. We random crop images to $3 \times 64 \times 64$ and scale to $[-1, 1]$.

We do require some images with ground truth for evaluation. We use images and labels from the ADE20K dataset (Zhou et al., 2016) for the kitchen and bedroom scene categories as the test set. For each scene category, we create a training dataset and validation dataset of randomly selected examples. For bedrooms, the training and validation each contain 640 examples. For kitchens, they each contain 320 examples. The sizes are limited by the number of annotations available in ADE20K for each scene category. We chose kitchens and bedrooms as they are the largest scene categories in the LSUN dataset and because we have a sufficient number of densely labeled examples in ADE20K.

For each identified object category in each scene, we create binary masks from the ADE20K dataset and pair them with their corresponding images. Due to the fact that ADE20K does not label behind occlusions, we combine labels to form the appropriate ground truth map. For example, pillows are often on the bed. We therefore define beds as the combination of beds, pillows, and comforters. For kitchen appliances, we define them as microwaves, ovens, dishwashers, stoves, and sinks. We evaluate the model versus baselines as pixel-wise binary classification. The mask represents the confidence of model that the pixel belongs to the specified object category. We run each experiment on a scene category and report the average precision as our metric.

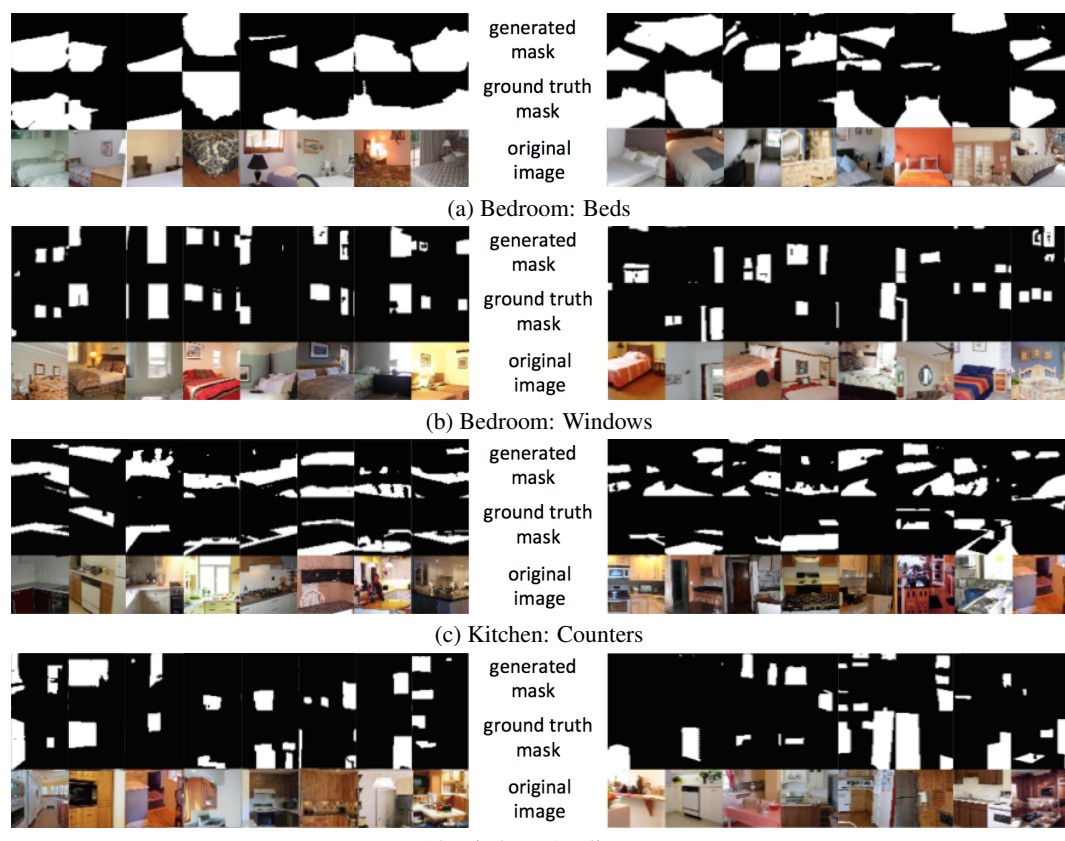

(a) Bedroom: Beds

(b) Bedroom: Windows

(c) Kitchen: Counters

(d) Kitchen: Appliances

Figure 4: **Example Results:** The left image contains some cases where we correctly segment the mask of objects, and the right image contains some failure cases. The first row is the generated mask, second row is the ground truth mask, and third row is the input image.

|  |  | Bedrooms | | Kitchens | |
|---|---|---|---|---|---|
|  |  | Bed | Window | Appliance | Counter |
| Unsupervised | Random | 0.31 | 0.13 | 0.10 | 0.07 |
|  | Autoencoder | 0.37 | 0.20 | 0.10 | 0.11 |
|  | Kmeans $1 \times 1$ | 0.34 | 0.17 | **0.15** | 0.08 |
|  | Kmeans $7 \times 7$ | 0.34 | 0.15 | 0.14 | 0.07 |
|  | Our Approach | **0.53** | **0.30** | 0.14 | **0.13** |
| Semi-Supervised | Average Mask | 0.52 | 0.19 | 0.12 | 0.10 |
|  | Random Init | 0.58 | 0.32 | 0.17 | 0.11 |
|  | Ours + Finetune | **0.71** | **0.51** | **0.21** | **0.19** |

Table 1: **Semantic Segmentation Average Precision:** We report average precision (area under precision-recall curve) on pixel-wise classification for four object categories. Our approach can segment images without supervision better than baselines. Moreover, the model can be fine-tuned with a little bit of labeled data to further improve results. We note that the unsupervised models still have access to scene labels.

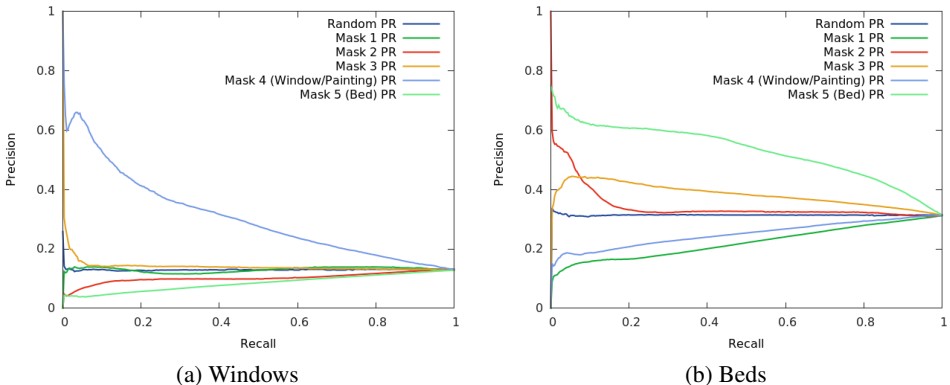

(a) Windows                                  (b) Beds

Figure 5: **Precision-Recall:** We plot precision-recall curves for each layer's mask. Our approach obtains good precision with low recall, suggesting that the model's most confident segmentations are fairly accurate. Notice how layers tend to specialize to certain object categories. The mask from layer 4 works well for segmenting windows, but the same layer does not work for beds. A similar trend exists for mask 5, but segments beds. This suggests that the model learns to group objects.

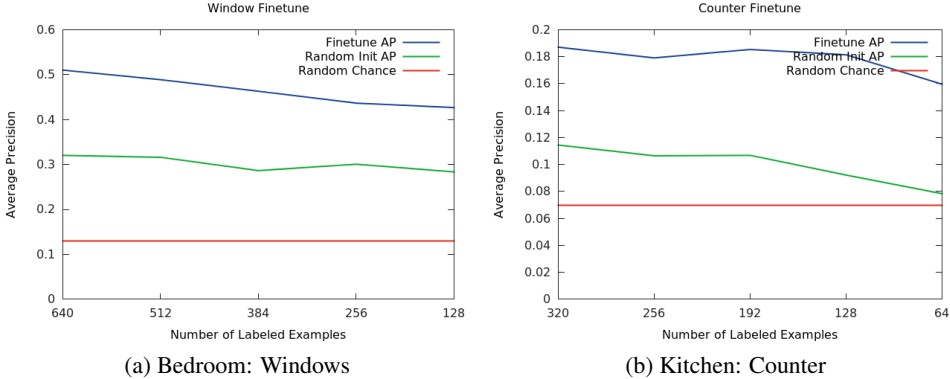

(a) Bedroom: Windows                           (b) Kitchen: Counter

Figure 6: **Performance versus size of labeled data.** We plot segmentation performance versus the size of the labeled dataset during fine-tuning. In general, more labeled data improves performance.

## 4.2 OBJECT SEGMENTATION

We quantitatively evaluate how well our model is able to do segmentation without pixel annotated training images in Table 1. Our results suggest that a decomposition of objects is automatically emerging in our model, which can segment objects better than unsupervised baselines. We note that unsupervised in our case refers to no pixel-level annotations. The models only have access to the scene class of each image.

For each scene category, we train on the LSUN dataset with 5 foreground layers. We extract the masks from each of the layers that are learned. We use the outputs of these masks as scores and calculate average precision as our metric. We compute the average precision for each layer on the isolated training set in ADE20K, and pick the layer that performs the best. We then report the average precision on the validation set. We also graph the precision-recall curves for the two objects for bedrooms in Figure 5.

Interestingly, each mask tends to capture a single object, suggesting the masks are learning a semantic decomposition. When evaluated on the bed objects, masks 5 performs the best, while mask 4 does worse than random. When evaluated on window objects, however, mask 4 does the best and mask 5 does worse than random.

We compare to a few baselines that also do not have access to pixel-level annotations. The random baseline corresponds to a random guess for each pixel. The autoencoder baseline corresponds to training the model with the composition probability set to 0 and with no permutations. In every case, our model with stochastic compositions receives a higher average precision, suggesting that removing and reordering layers does help to obtain an object segmentation. The kmeans baseline corresponds to clustering RGB patches across the dataset, and using distance to cluster centers as a segmentation score. We try both $1 \times 1$ and $7 \times 7$ patches with the same number of clusters as our model ($K = 5$). For each object category, we find the best performing cluster center on the ADE20K training set and evaluate with this cluster center on the validation set. In almost every case, our model outperforms this baseline.

Finally, we conduct an ablation on the model to understand why each layers learn to segment different objects. Normally in our experiments, each layer is initialized both randomly and independently. We tried initializing each stream to be the same (but not tying weights in training). We found that each stream tends to produce similar objects, and performance significantly drops (beds dropped to $0.41$ AP and windows dropped to $0.16$ AP). Since we also randomly permute foreground layers during each training iteration, this seems to effectively isolate initialization as a factor that leads to diverse outputs.

## 4.3 REFINING WITH LABELS

We can incorporate some pixel-level human labels into the model to improve performance with semi-supervised learning. After training the model without labels as before, we finetune masks on the ADE20K training set for each object class.

As an additional baseline, we calculate the average segmentation over the ADE20K dataset for each object category. Recall that even our model did not have access to this prior because it never saw a densely labeled image! For each object category we average each mask from the labeled dataset and we evaluate with this single mask.

The bottom rows of Table 1 shows that our approach provides a better initialization for supervised segmentation than baselines. For all object categories, the unlabeled model outperforms the average mask, suggesting the model outperforms naive priors estimated with labeled data, even though it never saw labels. For the bed objects, the simple prior comes close, possibly because beds are large and usually in a certain location (bottom). Windows, on the other hand, could be in many different locations and are usually smaller, hence the prior could not perform as well.

Figure 6 shows how performance changes with the size of the labeled data in our finetuning experiments. Interestingly, our supervised outperforms scratch initialization even with 20% of the training data. This shows that in the semi-supervised setting the model can be trained with much fewer examples. Another interesting note is that for the window object class our unsupervised model obtains performance close to the supervised random initialization model.

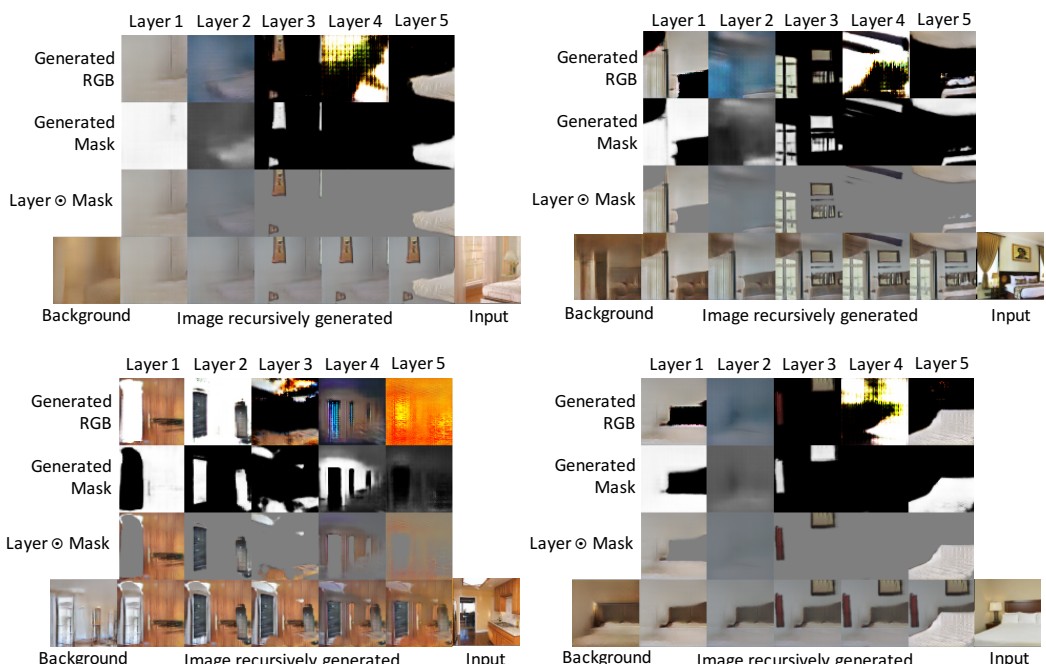

Figure 7: **Visualizing Layer Decomposition:** We visualize the outputs of different layers from our model given an input image. Different layers automatically emerge to both reconstruct objects and their masks, which we use for semantic segmentation. Moreover, this enables potential graphics applications, such as de-occluding objects in an image.

## 4.4 LAYER VISUALIZATION

We qualitatively show examples of images that are built up layer by layer in Figure 7. For each example we give the original image that was used as input, partial reconstructions for each layer that is added, the layer that is added, and the mask that the layer uses. These results suggest that as the generative model improves we will be able to remove layers that the model learns to see behind objects. For example, in the bottom right we can see that when the bed layer (layer 5) is removed we are able to uncover the headboard behind it.

The visualization highlights that our model learns a layer decomposition that allows counterfactual images to be realistic, however layers can emerge to capture other semantics besides objects. For example, some layers appear to capture lighting or textures in the scene. When these layers are removed, the resulting composition still looks realistic, but the lighting has changed. This suggests our approach can be useful for automatically learning semantic compositions of natural data, such as images or audio.

## 5 CONCLUSION

We propose a simple principle for learning object segmentation with data only labeled by scene. We capitalize on the observation that images are naturally compositional, and counterfactual images can provide a signal for learning to segment. Since annotating large-scale and pixel dense training data for segmentation is expensive, we believe developing approaches for segmentation requiring minimal labeling can have significant impact on many applications.

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
