# OpenReview forum: "Counterfactual Image Networks"
_ICLR.cc/2018/Conference — Reject_

### Official Review · AnonReviewer3 · 2017-11-24
**interesting idea and results, more experiments are needed**

**Rating:** 5
**Confidence:** 4

**Review:**

Paper summary: The paper proposes a generative model that decomposes images into multiple layers. The proposed approach is GAN-based, where the objective of the GAN is to distinguish real images from images formed by combining the layers. Some of the layers correspond to objects that are common in specific scene categories. The method has been tested on kitchen and bedroom scenes.

Paper Strengths:
+ The idea of the paper is interesting.
+ The learned masks for objects are neat.
+ The proposed method outperforms a number of simple baselines.

Paper Weaknesses:

- The evaluation of the model is not great: (1) It would be interesting to combine bedroom and kitchen images and train jointly to see what it learns. (2) It would be good to see how the performance changes for different number of layers. (3) Regarding the fine-tuning baselines, the comparison is a bit unfair since the proposed method performs pooling over images, while the baseline (average mask) is not translation invariant.

- It is unclear why "contiguous" masks are generated (e.g., in figure 4). Is there any constraint in the optimization? This should be explained in the rebuttal.

- The method should not be called "unsupervised" since it knows the label for the scene category. Also, it should not be called "semantic segmentation" since there is no semantics associated to the object. It is just a binary foreground/background mask.

- The plots in Figure 5 are a bit strange. The precision increases uniformly as the recall goes up, which is weird. It should be explained in the rebuttal why that happens.

- Similar to most GAN-based models, the generated images are not that appealing.

- The claim about object removal should be toned down. The method is not able to remove any object from a scene. Only, the learned layers can be removed.

---

### Official Review · AnonReviewer2 · 2017-11-27
**review: experiments insufficient**

**Rating:** 4
**Confidence:** 4

**Review:**

This paper proposes a neural network architecture around the idea of layered scene composition.  Training is cast in the generative adversarial framework; a subnetwork is reused to generate and compose (via an output mask) multiple image layers; the resulting image is fed to a discriminator.  An encoder is later trained to map real images into the space of latent codes for the generator, allowing the system to be applied to real image segmentation tasks.

The idea is interesting and different from established approaches to segmentation.  Visualization of learned layers for several scene types (Figures 3, 7) shows that the network does learn a reasonable compositional scene model.

Experiments evaluate the ability to port the model learned in an unsupervised manner to semantic segmentation tasks, using a limited amount of supervision for the end task.  However, the included experiments are not nearly sufficient to establish the effectiveness of the proposed method.  Only two scene types (bedroom, kitchen) and four object classes (bed, window, appliance, counter) are used for evaluation.  This is far below the norm for semantic segmentation work in computer vision.  How does the method work on established semantic segmentation datasets with many classes, such as PASCAL?  Even the ADE20K dataset, from which this paper samples, is substantially larger and has an established benchmarking methodology (see http://placeschallenge.csail.mit.edu/).

An additional problem is that performance is not compared to any external prior work.  Only simple baselines (eg autoencoder, kmeans) implemented by this paper are included.  The range of prior work on semantic segmentation is extensive.  How well does the approach compare to supervised CNNs on an established segmentation task?  Note that the proposed method need not necessarily outperform supervised approaches, but the reader should be provided with some idea of the size of the gap between this unsupervised method and the state-of-the-art supervised approach.

In summary, the proposed method may be promising, but far more experiments are needed.

---

### Official Review · AnonReviewer1 · 2017-11-28
**Unclear why it works the way it is advertised**

**Rating:** 4
**Confidence:** 4

**Review:**

This paper creates a layered representation in order to better learn segmentation from unlabeled images. It is well motivated, as Fig. 1 clearly shows the idea that if the segmentation was removed properly, the result would still be a natural image. However, the method itself as described in the paper leaves many questions about whether they can achieve the proposed goal.

I cannot see from the formulation why would this model work as it is advertised. The formulation (3-4) looks like a standard GAN, with some twist about measuring the GAN loss in the z space (this has been used in e.g. PPGN and CVAE-GAN). I don't see any term that would guarantee:

1) Each layer is a natural image. This was advertised in the paper, but the loss function is only on the final product G_K. The way it is written in the paper, the result of each layer does not need to go through a discriminator. Nothing seems to have been done to ensure that each layer outputs a natural image.

2) None of the layers is degenerate. There does not seem to be any constraint either regularizing the content in each layer, or preventing any layer to be non-degenerate.

3) The mask being contiguous. I don't see any term ensuring the mask being contiguous, I imagine normally without such terms doing such kinds of optimization would lead to a lot of fragmented small areas being considered as the mask.

The claim that this paper is for unsupervised semantic segmentation is overblown. A major problem is that when conducting experiments, all the images seem to be taken from a single category, this implicitly uses the label information of the category. In that regard, this cannot be viewed as an unsupervised algorithm.

Even with that, the results definitely looked too good to be true. I have a really difficult time believing why such a standard GAN optimization would not generate any of the aforementioned artifacts and would perform exactly as the authors advertised. Even if it does work as advertised, the utilization of implicit labels would make it subject to comparisons with a lot of weakly-supervised learning papers with far better results than shown in this paper. Hence I am pretty sure that this is not up to the standards of ICLR.

I have read the rebuttal and still not convinced. I don't think the authors managed to convince me that this method would work the way it's advertised. I also agree with Reviewer 2 that there is a lack of comparison against baselines.

---

### Author Response · Authors · 2018-01-05
**Response To Reviews**

Thank you for reading our paper! We are glad reviewers found our paper to be interesting.

Semantic segmentation models typically require large amounts of manually labeled images, which is expensive to collect. We instead develop a new method for learning to segment images without dense supervision. We propose a principle of “object removability” that we capitalize on to learn to segment images with unlabeled data. We believe this paper will have wide interest at the conference because it proposes a new signal for learning with weakly labeled data.

AnonReviewer1

“Each layer is a natural image.” We wish to clarify a misunderstanding. The loss function is on the final product G_K, but the final product at each iteration is formed by a randomly selected and permuted subset of the foreground layers. This is how we encode the idea of object removability; the generator must learn a set of foreground layers such that even with only a random subset of the layers included, G_K looks natural. Motivated by the example in Figure 1, we hypothesize that this will constrain the layers to learn a natural semantic representation (which our experiments suggest happens).

“None of the layers is degenerate.” Our model learns layers that generally represent an object category because objects can be removed from images. However, other features besides objects can be removed (such as lighting) and still produce a natural looking image. This happens in our model as well. For example, Figure 7 (bottom right image) shows that layer 2 has learned a blue lighting. We modified the text to discuss this.

“The mask being contiguous.” We do not require that the mask is contiguous, which is a strength of our method. For example, in Figure 7: Layer 2 on the bottom right and Layer 3 on the other three learn a non-contiguous mask because windows are not contiguous. On Layer 3 of the top right image, the mask learns several window-like objects, which it would not be able to do if we enforced a continuous constraint.

“The paper is not unsupervised.” We have modified the paper to tone down this claim.

AnonReviewer2

Evaluating on many scene classes: A limitation of GANs is that the generated images have limited variability. Our method uses a similar number of categories as state-of-the-art GANs. Scaling up GANs is orthogonal to this paper and out-of-scope. We believe the evaluation is suitable to show the efficacy of our approach.

Comparison to prior work: Thank you for this suggestion. We do train and show how we compare to a vanilla supervised CNN in Table 1 under the Random Init row.

AnonReviewer3

Evaluation of model. We could combine both the kitchen and bedroom images and jointly train, however current GAN objectives have a problem of mode collapse and are not able to capture the full variability of the dataset. Correcting mode collapse is out-of-scope.

There is no constraint in the model to enforce contiguous masks. See above in the response to AnonReviewer1.

We have modified the paper to tone down the claim that we can do unsupervised segmentation, and clarified that we do not attach semantics to the object. We also have modified the paper to tone down the object removal claim

In Figure 5, we plot how the precision-recall curves for each layer compare to randomly permuting pixels. We see that, as an example in (a), one layer has high precision in the low recall regions, suggesting that the pixels that were given the highest weights capture the “window” object well. When all the other layers are evaluated on the “window” object, they do poorly, with some masks doing worse than random on the low recall regions, suggesting that the pixels activated in the window layer tend not to be activated in the other layers.

---

### Decision · Program_Chairs · 2018-01-29
**ICLR 2018 Conference Acceptance Decision**

**Decision:**

Reject

**Comment:**

All reviewers acknowledge that the idea of the paper is interesting but have expressed serious concerns on empirical evaluations. The paper is not suitable for publication in its current form.